Diagnosis of hearing deficiency using EEG based AEP signals: CWT and improved-VGG16 pipeline

Islam Md Nahidul 1 nahidul76.edu@gmail.com
http://orcid.org/0000-0002-0625-2327 Sulaiman Norizam 1
Farid Fahmid Al 2
Uddin Jia 3
http://orcid.org/0000-0002-5507-9399 Alyami Salem A. 4
Rashid Mamunur 1
http://orcid.org/0000-0002-3094-5596 P.P. Abdul Majeed Anwar 5 6
http://orcid.org/0000-0003-0756-1006 Moni Mohammad Ali 7 m.moni@uq.edu.au
1 Faculty of Electrical and Electronics Engineering Technology, Universiti Malaysia Pahang , Pekan, Pahang , Malaysia
2 Faculty of Computing and Informatics, Multimedia University , Malaysia
3 Technology Studies Department, Endicott College, Woosong university , Daejeon , South Korea
4 Department of Mathematics and Statistics, Imam Mohammad Ibn Saud Islamic University , Riyadh , Saudi Arabia
5 Innovative Manufacturing, Mechatronics and Sports Laboratory, Faculty of Manufacturing and Mechatronic Engineering Technology, Universiti Malaysia Pahang , Pekan, Pahang , Malaysia
6 Centre for Software Development & Integrated Computing, Universiti Malaysia Pahang , Pekan, Pahang , Malaysia
7 School of Health and Rehabilitation Sciences, Faculty of Health and Behavioural Sciences, The University of Queensland St Lucia , Australia
Liu Pengcheng
Electronic publication date: 2021 Sep 29
Publication date: 2021
Volume: 7
Electronic Location ID: e638
Received 2021 Apr 1; Accepted 2021 Jun 21
Copyright: © 2021 Islam et al.
Copyright year: 2021
Copyright holder: Islam et al.
License: This is an open access article distributed under the terms of the Creative Commons Attribution License, which permits unrestricted use, distribution, reproduction and adaptation in any medium and for any purpose provided that it is properly attributed. For attribution, the original author(s), title, publication source (PeerJ Computer Science) and either DOI or URL of the article must be cited.
License URL: https://creativecommons.org/licenses/by/4.0/

Keywords: Electroencephalogram, Deep learning, Auditory Evoked potential, Transfer learning, VGG16

Funding: Ministry of Higher Education FRGS/1/2018/TK04/UMP/02/3 Universiti Malaysia Pahang RDU190109 The authors would like to thank the Ministry of Higher Education for providing financial support under Fundamental research grant No. FRGS/1/2018/TK04/UMP/02/3 (University reference RDU190109) and Universiti Malaysia Pahang for laboratory facilities as well as additional financial support under Internal Research grant RDU190109. The funders had no role in study design, data collection and analysis, decision to publish, or preparation of the manuscript.

==============================
Hearing deficiency is the world’s most common sensation of impairment and impedes human communication and learning. Early and precise hearing diagnosis using electroencephalogram (EEG) is referred to as the optimum strategy to deal with this issue. Among a wide range of EEG control signals, the most relevant modality for hearing loss diagnosis is auditory evoked potential (AEP) which is produced in the brain’s cortex area through an auditory stimulus. This study aims to develop a robust intelligent auditory sensation system utilizing a pre-train deep learning framework by analyzing and evaluating the functional reliability of the hearing based on the AEP response. First, the raw AEP data is transformed into time-frequency images through the wavelet transformation. Then, lower-level functionality is eliminated using a pre-trained network. Here, an improved-VGG16 architecture has been designed based on removing some convolutional layers and adding new layers in the fully connected block. Subsequently, the higher levels of the neural network architecture are fine-tuned using the labelled time-frequency images. Finally, the proposed method’s performance has been validated by a reputed publicly available AEP dataset, recorded from sixteen subjects when they have heard specific auditory stimuli in the left or right ear. The proposed method outperforms the state-of-art studies by improving the classification accuracy to 96.87% (from 57.375%), which indicates that the proposed improved-VGG16 architecture can significantly deal with AEP response in early hearing loss diagnosis.

Introduction

Hearing deficiency is the widespread form of human sensory disability; it is the partial or complete inability to listen to the ear’s sound. The world health organization (WHO) reports that 466 million people were living with hearing loss in 2018, projected to exceed 630 million by 2030 and more than 900 million by 2050 (World Health Organization (WHO), 2021). An early and effective hearing screening test is essential for address the vast population concern. That helps to reduce the hearing deficiency by taking necessary steps at an appropriate time. Conventional listening tests and audiograms appear to be subjective assessments that significantly demand medical and health services. The audiogram reflects the hearing threshold across the speech frequency spectrum, usually between 125 and 8,000 Hz. The traditional hearing impairment testing technique is very time-consuming, takes sufficient clinical time and expertise to interpret and maintain since it requires the person to respond directly. In the application of hearing aid, other issues, such as hearing loss’s consequence (Holmes, Kitterick & Summerfield, 2017), the circumstances of the auditory stimulus (such as the background noise of the stimulus, locations of the stimulus (Das, Bertrand & Francart, 2018; Das et al., 2016) ), attention-altering techniques is still an open question.

Various hearing impairment testing techniques have been conducted to address these issues, and among them, EEG-based auditory evoked potentials (AEPs) are most widely used (Zhang et al., 2006; Mahmud et al., 2019). Nowadays, the classification of AEP signal is most commonly used in many brain-computer interface (BCI) applications (Gao, Wang & Gao, 2014) and brain hearing issues (Sriraam, 2012). In fact, the AEP signal is widely used to recognize hearing capability, assessment, and neurological hearing impairment identification. The AEP signals are reflected by the brain’s electrical activity changes in the body’s sensory mechanisms in response to the auditory stimulus. The diagnosis of hearing loss typically involves four main stages: acquisition of data, data pre-processing, feature extraction and selection, and classification. The feature extraction is traditionally conducted by analyzing the time-domain, frequency-domain, and time-frequency domain techniques, which help to extract the information from the original raw data. The extracted features are then used as an input to the machine learning or deep learning models for training. However, traditional diagnosis methods have some drawbacks. For example, traditional hearing loss approaches are often based on manual feature selection. As a consequence, if the manually chosen features are ineffective for this task, the hearing loss recognition performance will decrease considerably. Furthermore, handcrafted features for different classification tasks are task-specific, meaning that features that render predictions correctly are not acceptable under certain conditions for other scenarios (Acir, Erkan & Bahtiyar, 2013; Acir, Özdamar & Güzeliş, 2006).

Although the researchers have employed a wide range of machine learning and deep learning algorithms, recognizing the most effective classifier is still an open question. Among machine learning-based classifiers, support vector machine (SVM) (Mahmud et al., 2019), k-nearest neighbors (k-NN) (Thorpe & Dussard, 2018; Rashid et al., 2021), artificial neural network (ANN) (Mccullagh et al., 1996), linear discriminant analysis (LDA) (Grent-‘t-Jong et al., 2021) Naïve Bayesian (NB) (Shirzhiyan et al., 2019) are widely used in neurological response classification. Nowadays, the convolutional neural networks (CNNs) are the most preferred approach in the different classification tasks, particularly in image classification (Lecun, Bengio & Hinton, 2015). In some recent studies, CNNs have shown promising performances in EEG signal classification: in seizure detection (Ansari et al., 2019), depression detection (Liu et al., 2018), and sleep stage classification (Ansari et al., 2018). Ciccarelli et al. (2019) proposed a novel architecture of the neural network and showed that their approach outperforms the linear methods in decision windows of 10s. They have used eleven subjects in the experiment: with the wet EEG, the decoding accuracy was improved from 66% to 81%, and with the dry EEG, the decoding accuracy was improved from 59% to 87%. McKearney & MacKinnon (2019) used a deep neural network approach to classify paired auditory brainstem responses. They used 232 paired ABR waveforms (190 paired ABR waveforms for training the model and 42 paired waveforms for performance evaluation) from eight normal hearing subjects and achieved 92.9% testing accuracy. Although they achieved an excellent performance to identify the auditory brainstem response, the testing set is too small, and more dataset is needed to test the model performance. Mccullagh et al. (1996) reported a 73.7% accuracy using the artificial neural network to classify 166 auditory brainstem responses (ABRs) with 2,000 repetitions. Ibrahim, Ting & Moghavvemi (2019) used multiple classification techniques for detecting the hearing condition; the SVM algorithm outperforms the other algorithms by achieving a classification accuracy of 90%. They used a nonlinear feature extraction method to extract adequate information from the AEP signals. Dietl & Weiss (2004) evaluated an application to achieve detection of frequency-specific hearing loss where they used the wavelet packet transform (WPT) as a feature extraction method and support vector machines (SVM) classifier to transient evoked otoacoustic emissions (TEOAE). They achieved a maximum of 74.7% accuracy with the testing dataset. Nonetheless, the overall accuracy is not favourable enough to be utilized in real-life applications. Tang & Lee (2019) proposed a novel hearing deficiency diagnosis method using three-level wavelet entropy, followed by MLP, trained by hybrid Tabu search-Particle Swarm Optimization (TS-PSO). Their approach achieved 86.17% testing accuracy; it still needs improvement for real-time applications. Sanjay et al. (2020) used machine learning approaches for human auditory threshold prediction. The absolute threshold test (ATT) method was used for feature extraction from the auditory signals. The extracted feature was then classified using multiple classification methods. Among all the classification methods, a maximum of 93.94% accuracy was achieved with the SVM classifier. Xue et al. (2018) used participants’ articulatory movements with or without hearing impairment during nasal finals for hearing impairment diagnosis. Six different kinematic features: standard deviation of velocity, minimum velocity, maximum velocity, mean velocity, duration, displacement was used to extract the information from the hearing impairment (HI) patient and normal hearing (NH) participants. The classification was conducted with a support vector machine, radial basis function network, random forest, and C4.5. The maximum accuracy was 87.12% using a random forest classifier via (displacement and duration feature). Zhang et al. (2006) proposed an auditory brainstem response classification method. They used wavelet analysis for feature extraction and Bayesian networks to classify the auditory responses. Discrete wavelets transform (DWT) was used to extract the time-frequency information from the raw signals. A maximum of 78.80% testing accuracy was achieved in their proposed approach; it needs more improvement in testing accuracy.

The emphasis in our study is on a concise decision window. However, a concise window contains less information and more difficult to achieve high performance but provide an effective solution for early detection of hearing disorder. The short decision window is considered one of the prerequisites to develop the real-life application, but limited studies have been carried out to investigate this issue (Deckers et al., 2018). Moreover, selecting a short decision window makes the system faster by reducing the computational complexity of the system. On the other hand, Deep learning (DL) approaches can provide an effective solution because of their effective feature learning capability to overcome the above limitations (Krizhevsky, Sutskever & Hinton, 2017; Nossier et al., 2019; Shao et al., 2019; Bari et al., 2021; Mahendra Kumar et al., 2021). Deep learning models have several hidden layers that can explicitly learn hierarchical representations. From model training, deep architectures can select discriminatory representations, which are helpful for precise predictions according to the training data in subsequent classification stages. Although the DL models have successful application in hearing loss diagnosis tasks, there are still some issues with DL approaches. A few investigations (Ciccarelli et al., 2019; McKearney & MacKinnon, 2019) have been conducted using deep models with more than 10 hidden layers for hearing loss diagnosis. A large number of labelled data and computations resources are typically required during the training model from scratch. In the proposed study, we used the transfer learning (TL) method to address the challenges of training a deep model from scratch. The TL method is used to expedite the deep learning model training phase and effectively learns the hierarchical representations. The process is accomplished by using the pre-trained TL method that has been pre-trained on vast datasets of natural images. The proposed pre-trained model provides the lower-level weights for the target neural network, while the higher-level weights are fine-tuned for the hearing deficiency diagnosis task. Consequently, the proposed TL method offers a rational initialization for the target model and decreases the number of model’s parameters. In this manner, TL significantly enhances the performance of the training process. Here, we summarized the main contribution of this paper. We have presented a hearing deficiency identification system based on deep CNN, where a transfer learning strategy has been used to improve the training process. To fit the AEP dataset in our model, we fine-tune the high-level parameters, consisting of unfreezing some part of the pre-training model and re-training it. The lower-level parameters are transferred from the previous trained deep architecture.

In the proposed approach, we also changed some high-level parameters, reduced the number of parameters and complexity of the TL architecture, which helps in improving the performance of the VGG16 model for our dataset and reduces the computational time of the training process.

The experiment is conducted in a short decision window (1s and 2s), minimizing the impact of additional features and reducing time consumption, which shows the proposed system robustness and applicability in real-life application.

The rest of the manuscript is arranged as follows: a detailed data description, data pre-processing, and the transformation process of CWT are implemented in the Materials and Methodology section. A detailed description of the development of the proposed pre-trained model and fine-tuning procedure for hearing deficiency diagnosis is also described in this section. Experimental performance to determine the models’ validation is described in the Result of the Experiment and Analysis section. The Discussion section exhibits a discussion on the comparison of the proposed model with related studies, along with the key advantages of our proposed method over the previous studies. The Conclusion section represents the outcome of the present study.

Materials and methodology

The aim of this study is to build an intelligent auditory sensation system for hearing loss diagnosis with high performance. The overall procedure of the proposed hearing loss diagnosis method is demonstrated in Fig. 1. The proposed framework consists of few steps, including data collection, pre-processing, time-frequency analysis, and building a pre-trained model with fine-tuning. We have used a publicly available online dataset in the data collection phase instead of data collection ourselves. We converted the raw signal into a time-frequency image using continuous wavelet transform (CWT). Then, the proposed deep CNN (improved-VGG16) method is applied in the time-frequency images for diagnosis the hearing loss. In the TL model, the pre-trained ImageNet dataset has been used, and the size of the images is 224 * 224 pixels in RGB. The entire dataset has been converted into a time-frequency image after data collection and resized in height-224 * width-224 * depth-3. The VGG16 uses natural images which are different from the time-frequency images of AEP. So, to fit the AEP dataset in the TL model, we replaced some VGG16 layers with the new layers and then fine-tuned the improved VGG16 model.

Figure 1 The overall procedure of hearing deficiency diagnosis method.

Data description

Experimental AEP datasets are provided by ExpORL, Dept. Neurosciences, KULeuven, and Dept. Electrical Engineering (ESAT), KULeuven (Das, Francart & Bertrand, 2020). A 64-channel BioSemi Active Two system was used for recording the AEP data, which was 8,196 Hz sampling rate. The entire data was collected from 16 normal-hearing subjects, and the trial was repeated 20 times from each subject. The recordings were conducted in a soundproof, electromagnetically shielded space. The auditory stimuli were presented at 60 dBA by Etymotic ER3 insert earphones and were low-pass filtered with a cut-off frequency of 4 kHz. As simulation software, APEX 3 was used (Francart, van Wieringen & Wouters, 2008). Three male Flemish speakers narrated four Dutch stories as auditory stimulation (Radioboeken, 2021). Every story lasted 12 min and was divided into two segments of 6 min each. Silent segments that lasted more than 500 ms were shortened to 500 ms. The stimuli were equal in root-mean-square intensity and perceived as equally loud. The experiment was divided into eight sections, each lasting six minutes. Subjects were presented with two parts of two storylines in each trial. The left received one part, while the right ear received the other part. To prevent the lateralization bias described by Das et al. (2016), the attended ear was alternated over successive trials to ensure that each ear received an equal volume of data. Each subject received stimuli in the same order, either dichotically or after head-related transfer function (HRTF) filtering (simulating sound coming from ±90°). As with the attended ear, the HRTF/dichotic condition was randomized and balanced within and over subjects.

Data preprocessing

The pre-processing of the AEP data is the first phase after data collection. In this study, the trials were filtered with a high pass (0.5 Hz cut off) and downsampled from the sampling rate of 8,192 Hz to 128 Hz. Here, we have investigated sixteen subjects, and each trial has been segmented into the same length. The entire dataset has been segmented into short decision windows (1s and 2s) and considered each decision window an observation. The straightforward reason to select the concise decision windows is to reduce the computational complexity and make the system faster, which will help detect the early hearing disorder. From each subject, 200 observations have been picked, and finally, we achieved a total of 3,200 observations. After data filtering and window selection, the AEP data of subject-1, channel-1 in the time domain, is shown in Fig. 2 when the subject hears auditory stimulus through headphones defined as left and right labels.

Figure 2 AEP raw data plotting in 2s decision window: (A) hear auditory stimulus with the left ear (B) hear auditory stimulus with the right ear.

CWT for time-frequency analysis

CWT is a time-frequency feature extraction approach that offers multi-scale signal refinement by scaling and translating operations. After the data pre-processing step, the segmented dataset transforms from the time domain to the time-frequency domain using the CWT.

The CWT can automatically adapt the time-frequency signal analysis criteria and clearly explain the signal frequency change with time (Yan, Gao & Chen, 2014). The CWT is widely used for feature extraction and can be considered a mathematical tool for transforming time-series into a different feature space. This study uses CWT as a feature extraction method that converts the raw signal into 2-D time-frequency images from 1-D time-domain signals. An internal signal operation and a series of wavelets are performed by the wavelet transforms. The mother wavelet is scaled and translated to create the wavelet set, which is a family of wavelets ψ(t), shown as

(1) ψS,τ(t)=1Sψ(t−τS)

Here, S represents the scale parameter inversely related to frequency, and τ represents the translation parameter.

The signal x(t) can be achieved by a complex conjugate convolution operation, mathematically defined as follows (Huang & Wang, 2018):

(2) W(s,τ)=x(t),ψS,τ=1s∫x(t)ψ∗(t−τS)dt

where Ψ∗(⋅) denotes the complex conjugate of the above function Ψ(⋅) and This operation decomposes the signal x(t) in a series of wavelet coefficients, in which the base function is the wavelet family. In the equation, the s and τ are two types of parameters in the family wavelets. The signal x(t) is transformed and projected to the time and scale dimensions of the family wavelets.

In this study, we use wavelet basis functions (Mother Wavelets). The time-frequency images are then used as the input of the proposed TL model. The transformation process of CWT is shown in Fig. 3.

Figure 3 The transformation process from time-domain signal to time-frequency domain image.

Finally, we concatenate the 64 channels data in (M*M) for preparing an observation, where the value of M is set to eight. So, each observation provides the time-frequency information of 64 channels. Figure 4 shows the time-frequency image of 64 channels.

Figure 4 The time-frequency image of 64 channels data.

Hearing deficiency diagnosis using deep TL

The proposed system presented a deep TL method based on improved-VGG16 architecture for hearing loss diagnosis. The VGG16 uses natural images which are different from the time-frequency images of AEP. The improvement consists of replacing some VGG16 layers with the new layers and then fine-tuning the layers to fit the time-frequency AEP dataset in the model.

Convolutional neural network architecture

LeCun et al. (1998) proposed the convolutional neural networks (CNN), one of the best pattern recognition methods. The locally trained filters are used in this system to extract the visual features through the input image. CNN’s internal layer structure consists of a convolution layer, pooling layer, and fully connected layer. The complete procedure of CNN is shown in Fig. 5.

Figure 5 Typical convolutional neural network architecture.

Convolution layer

The convolutional operations provide the more advanced feature representation. Several fixed-size filters allow the complex functions to be used in the input image (Ravi et al., 2017). The same weights and bias values are used in the whole image in each filter. This technique is called the weight-sharing mechanism, and it makes it possible to represent the entire image with the same characteristic. A neuron’s local receptive field reflects the neuron’s region in the previous layer. This study uses the ‘ReLU activation function (Alpaydin, 2021). Let c × c is the size of the kernel or filter, and i represent the time-frequency image. The weight and bias of the filter are denoted by w and b, respectively. The output O0,0 can be computed using Eq. (3), where f denotes the activation function. This study used the ReLU activation function. In most of the classification tasks, the ReLU activation function has demonstrated superior performance in terms of accelerating convergence and mitigating the issue of vanishing gradients (Krizhevsky, Sutskever & Hinton, 2017). The mathematical representation of the ReLU activation function can also be seen in Eq. (4),

(3) O0,0=f(b+∑t=0c⁡∑r=0c⁡wt,ri0+t,0+r)

(4) f(x)={xx>00else.

Pooling Layer

The pooling method is used in the feature maps, which have gone through convolution and activation function. The pooling layer computes the local average or maximum value, reducing the complexity and retaining the essential features, thus enhancing feature extraction performance. Fully connected layer

The convolutional and pooling layers alternately transfer the image features; after that, the fully connected layer received the image feature as an input. One or more hidden layers may have in the fully connected layer. By the data from the previous layer, each neuron multiplies the connection weights and adds a bias value. Before transmission to the next layer, the measured value is passed via the activation function. Eq. (5). displays neuronal calculations in this layer.

(5) fc1=f(b+∑q=1M⁡w1,q∗Oq)

where f is the activation function, w is the weight vector, O is the input vector of the qth neuron, and b is the bias value. SoftMax

The SoftMax activation function variates the logistic regression adapted to multiple classes and used in the output layer for classification purposes. It can be determined by Eq. (6) (Sermanet et al., 2013),

(6) classj=exp(sfj)∑q⁡exp(sfq)

Proposed pretrained model building and fine-tuning

In the convolutional neural network, the convolutional layers are used to extract the features from the dataset in a different manner, whereas the fully connected layers are used to classify the extracted features. The most forthright approach for enhancing the feature learning capability is to increase the depth or width of the deep neural network. However, this can lead to two issues: the first concern is that a deeper or wider model typically has more parameters, rendering the expanded network more vulnerable to overfitting. The second concern is that it raises the use of computing resources substantially.

To overcome these flowing issues and extract the AEP feature efficiently, the VGG16 network utilizes several parallel layers with different convolutional kernel sizes. It concatenates the outputs at the end of the pre-trained network. In the proposed TL model, we replace some layers of VGG16 with the new layers to fit the AEP dataset in the pre-trained network, which enhances hearing loss identification performance. The replacement process consists of adding some dense layers in the fully connected block of VGG16 architecture and adding the dropout layers after every dense layer. A densely connect layer learns features from all the previous layer’s features. The dense layer performs a matrix-vector multiplication, and with the help of backpropagation, the parameters can be trained and updated. The dense layer is used to change the vector’s dimensions and applies in other operations like rotation, scaling, and translation. Mele & Altarelli (1993) reported that on the CIFAT-10 dataset, the error rate 16.6% when testing the dataset in a convolutional neural network. They improved the model’s performance with an error rate of 15.6% when the dropout layer was utilized in the last hidden layer. We add the dropout layer after every dense layer in the fully connected block to reduce the model complexity and prevent overfitting. The neuron is temporarily dropped with the probability p at each iteration. Then, at every training step, the dropped-out neuron is resampled with the probability p, and a dropped-out neuron will be active at the next step. Here, the hyperparameter p is the dropout rate. Since the VGG16 uses the ‘ImageNet’ weight, which is trained with the natural image, and the proposed time-frequency images are not similar, more layers need to be fine-tuned where the weight is updated with the ‘ImageNet’ weight. This process helps to fit the time-frequency images with the TL architecture. The proposed fine-tuning consists of unfreezing some pre-trained network layers and re-train with the AEP dataset.

In the proposed approach, at first, we remove all the layers of VGG16 after the first 3 × 3 convolution layer of convolutional block five, as shown in Fig. 6, and replace the fully connected block there. Then, we add multiple dense layers at the end of the VGG16 model, and after every dense layer, we add the dropout layer. In the case of CNN, the convolutional layers extract the feature from the dataset, whereas the fully connected layers try to classify the extracted features. Consequently, adding more layers to the dense section can empower the network’s robustness and improve classification accuracy. So, despite using the two dense layers of the VGG16, here, we add three new dense layers units of 1,024, 512, and 288 in the fully connected block. Then, we add a dropout layer after each dense layer, and the dropout value is set to 0.2, 0.4, and 0.6, respectively. The reason behind adding the dropout layers is that the deep learning model reduces the performance due to overfitting, and the dropout layers reduce the model complexity and prevent overfitting. These techniques help in enhancing the performance in the hearing loss diagnosis. We also remove the top layer and adding a SoftMax layer (output layer) based on the targeted class. Based on the hyperparameters tuning technique, the proposed approach uses the ‘Adam’ optimizer to adjust the network weight with the batch size 64, and the learning rate is set to 0.0001. The parameters selection is made with the help of the ‘Keras-Tuner’ library. This library helps to select the most optimal set of hyperparameters for our architecture. Hyperparameters are the variable that governs the training process of the DL model and structure. There are two types of hyperparameters: first, model hyperparameters that help in selecting the number and width of the multiple hidden layers. Second, algorithm hyperparameters help to influence the speed and quality of the learning algorithm. All the hyperparameters selected to build the proposed architecture are based on ten different runs of the model. The following steps are used to train the model for hearing loss identification, shown in Box 1.

Figure 6 Transfer learning procedure of the proposed method.

Box 1 Training procedure of proposed TL architecture.

The training steps of the proposed TL architecture:	
Step 1:	Load the VGG16 base model with the pre-trained weights.	
Step 2:	Freeze some layers in the base model by setting trainable = False. In the nontrainable layers, the weights will not train.	
Step 3:	Create a new model by replacing some layers of VGG16 with new layers and retrain the layers with the layers where the trainable = True.	
Step 4:	Train the new model with the dataset.	

The detailed information of the parameter of the proposed TL architecture is shown in Table 1. Here, C means the targeted class.

Table 1 Parameter of proposed TL architecture.

Layer (type)	Output	Number of parameters	
Input	224 * 224 * 3	0	
Block1-Conv2D	224 * 224 * 64	1,792	
Block1-Conv2D	224 * 224 * 64	36,928	
Block1-MaxPooling2D	112 * 112 * 64	0	
Block2-Conv2D	112 * 112 * 128	73,856	
Block2-Conv2D	112 * 112 * 128	147,584	
Block2-MaxPooling2D	56 * 56 * 128	0	
Block3-Conv2D	56 * 56 * 256	295,168	
Block3-Conv2D	56 * 56 * 256	590,080	
Block3-Conv2D	56 * 56 * 256	590,080	
Block3-MaxPooling2D	28 * 28 * 256	0	
Block4-Conv2D	28 * 28 * 512	1,180,160	
Block4-Conv2D	28 * 28 * 512	2,359,808	
Block4-Conv2D	28 * 28 * 512	2,359,808	
Block4-MaxPooling2D	14 * 14 * 512	0	
Block5-Conv2D	14 * 14 * 512	2,359,808	
Flatten-Flatten	1 * 1 * 100352	0	
fc1-Dense	1*1*1024	102,761,472	
dropout-Dropout	1*1*1024	0	
fc2-Dense	1*1*512	524,800	
dropout_1-Dropout	1*1*512	0	
Fc3-Dense	1*1*288	147,744	
dropout_2-Dropout	1*1*288	0	
Output-Dense	C	288*C+C	

During the training process, all the layers before convolutional block four are frozen. The weights are updated in the trainable layers, which helps in minimizing the errors between the predicted labels and the actual labels. The complete architecture of the proposed TL has demonstrated in Fig. 6.

Result of the experiment and analysis

This section represents the proposed hearing loss diagnosis method’s performance based on CWT and deep CNN architecture (improved-VGG16). First, we converted the time domain signal to time-frequency domain images. Then, the images are resized into height-224 * width-224 * depth-3, which is the suitable size of the proposed model. In this study, two different decision windows were tested: 1s and 2s. This term refers to the quantity of data required to make a single left/right decision. The practical reason behind selecting the shorter decision window is to detect the hearing condition quickly. The entire dataset was randomly split into the training set and testing set. Here, we used 70% dataset to train the architecture, and the rest of the dataset was used to test the model’s validation. This experiment has conducted with sixteen subjects where the subjects hear the auditory track. Based on listening to the auditory track with the ear, the dataset has been divided into two classes. The ‘Class1’ means the subject hears the auditory track with the left ear and the ‘Class2’ means the subject hears the auditory track with the right ear. With the (1s and 2s) decision windows, we randomly selected 200 observations from each subject. A total of 2,240 observations has been used for training the model and 960 observations for testing the performance.

For 1s window length, the performance of the proposed approach for each subject in terms of accuracy, precision, recall, f1-score and cohen’s kappa of all subjects is demonstrated in Table 2.

Table 2 Performance of proposed model for 1s decision window.

Subject	Accuracy	Precision	Recall	F1 Score	Cohens
Kappa	
Subject-1	0.9833	0.9688	1.0	0.9841	0.9666	
Subject-2	0.9667	1.0	0.9355	0.96667	0.9334	
Subject-3	0.8667	0.8108	0.9677	0.8824	0.7312	
Subject-4	0.9667	0.9393	1.0	0.9688	0.9331	
Subject-5	1.0	1.0	1.0	1.0	1.0	
Subject-6	0.8333	0.8387	0.8387	0.8387	0.6663	
Subject-7	1.0	1.0	1.0	1.0	1.0	
Subject-8	0.95	0.9667	0.9355	0.9508	0.9	
Subject-9	0.7667	0.7167	0.7933	0.7367	0.4833	
Subject-10	0.9833	1.0	0.9677	0.9836	0.9667	
Subject-11	0.8167	0.7409	1.0	0.8578	0.6241	
Subject-12	0.9833	1.0	0.9632	0.9853	0.9567	
Subject-13	0.7833	0.8214	0.7419	0.7797	0.5676	
Subject-14	0.76667	0.7453	0.8365	0.7892	0.5378	
Subject-15	0.9833	0.9688	1.0	0.9841	0.9666	
Subject-16	1.0	1.0	1.0	1.0	1.0	
Average ± SD	91.56% ± 8.91%	90.74% ± 10.47%	93.63% ± 8.25%	91.92% ± 8.79%	82.71% ± 18.34%	

Table 2 illustrates that in the case of subject-5, subject-7, and subject-16, our network achieves an unprecedented performance of 100%. Except for six subjects (Subjects-3, 6, 9, 11, 13 and 14), all subjects have achieved more than 90% accuracy. However, comparatively lower classification accuracy has been noticed by Subjects-3 (86.67%), Subject-6 (83.33%), Subject-9 (76.67%), Subject-11 (81.67%), Subject-13 (78.33%), and Subject-14 (76.67%). In the case of 1s decision window length, the average classification accuracy is 91.56%, whereas the standard deviation is 8.91%. Besides classification accuracy, other performance evaluation techniques (such as precision, recall, f1-score, and cohen kappa score) are also calculated to check the proposed model’s acuity. The average value of precision, recall, f1-score, and cohen kappa for sixteen subjects are 90.74%, 93.63%, 91.92%, 82.71%, respectively, whereas standard deviations are 10.47%, 8.25%, 8.79%, 18.34%, respectively. Figure 7 shows the overall accuracy and loss curve of the proposed TL method for the 1s decision window.

Figure 7 The overall accuracy and loss curve of the proposed TL method for 1s decision window.

For 2s window length, the performance of the proposed architecture is illustrated in Table 3. In this case, a maximum of 100% accuracy has achieved for subject-6, subject-7, subject-10, subject-16. Here, in the case of subject-16, we achieved 1.67% more accuracy compared to the 1s time window analysis. However, the proposed architecture achieves an unprecedented improvement (more than or equal to 90% for decision windows of 2s) in each subject. The lowest accuracy of 90% has been obtained in subject-13.

Table 3 Performance of proposed model for 2s decision window.

Subject	Accuracy	Precision	Recall	F1 Score	Cohens Kappa	
Subject-1	0.9833	1.0	0.9677	0.9836	0.9666	
Subject-2	0.9666	0.9393	1.0	0.9687	0.9331	
Subject-3	0.95	0.9666	0.9354	0.9508	0.9	
Subject-4	0.9666	0.9393	1.0	0.96875	0.9331	
Subject-5	0.9833	1.0	0.9677	0.9836	0.9666	
Subject-6	1.0	1.0	1.0	1.0	1.0	
Subject-7	1.0	1.0	1.0	1.0	1.0	
Subject-8	0.95	0.9375	0.9677	0.9523	0.8997	
Subject-9	0.9666	0.9677	0.9677	0.9677	0.9332	
Subject-10	1.0	1.0	1.0	1.0	1.0	
Subject-11	0.9333	0.9354	0.9354	0.9354	0.8665	
Subject-12	0.95	0.9666	0.9354	0.9508	0.9	
Subject-13	0.9	0.8787	0.9354	0.9062	0.7993	
Subject-14	0.9666	0.9393	1.0	0.9687	0.9331	
Subject-15	0.9833	0.9687	1.0	0.9841	0.9665	
Subject-16	1.0	1.0	1.0	1.0	1.0	
Average ± SD	96.87% ± 2.78%	96.49% ± 3.5%	97.57% ± 2.76%	97% ± 2.64%	93.73% ± 5.57%	

With the 2s decision window, the average value of accuracy precision, recall, f1-score, and cohen kappa for sixteen subjects are 96.87%, 96.49%, 97.57%, 97% and 93.73%, respectively. On the other hand, the standard deviation of precision, recall, f1-score, and cohen kappa are 2.78%, 3.50%, 2.76%, 2.64% and 5.57%, respectively. Figure 8 shows the overall accuracy and loss curve of the proposed TL method.

Figure 8 The overall accuracy and loss curve of the proposed TL method for 2s decision window.

To illustrate the performance of the proposed TL model in depth, the confusion matrix of all subjects has been given separately. A confusion matrix can be used to estimate the classification accuracy of a model visually. Figure 9 represent the confusion matrix with 1s decision windows analysis, whereas Fig. 10 represent the confusion matrix with 2s decision window analysis. In both figures, the letter A to P denotes the confusion matrix of subject-1 to subject-16, respectively.

Figure 9 Confusion matrix for 1s decision windows: (A) subject-1, (B) subject-2, (C) subject-3, (D) subject-4, (E) subject-5, (F) subject-6, (G) subject-7, (H) subject-8, (I) subject-9, (J) subject-10, (K) subject-11, (L) subject-12, (M) subject-13, (N) subject-14, (O) subject-15, (P) subject-16.

Figure 10 Confusion matrix for 2s decision windows: (A) subject-1, (B) subject-2, (C) subject-3, (D) subject-4, (E) subject-5, (F) subject-6, (G) subject-7, (H) subject-8, (I) subject-9, (J) subject-10, (K) subject-11, (L) subject-12, (M) subject-13, (N) subject-14, (O) subject-15, (P) subject-16.

The correct predictions are on the diagonal in the confusion matrix, while the incorrect predictions are off the diagonal. For example, in the case of Fig. 10A that denotes subject-1, a total of 59 observations (29 observations for class1, 30 observations for class2) have been recognized accurately among 60 observations. In both decision windows, the total testing set for sixteen subjects consists of 960 observations, in which 464 observations are in 'Class1', and 496 observations are in 'Class2'. For 1s decision windows, our network correctly detects 876 observations whilst 84 observations have been misclassified (shown in Fig. 9). On the other hand, for 2s decision windows, 930 observations have been accurately detected, whereas only 30 observations have been misclassified (shown in Fig. 10). Therefore, 2s decision windows provide a significant performance compared to the 1s decision windows.

Furthermore, to study the relationship between window length and detection performance, this study includes a comparison. Figure 11 visualizes the average performance of two decision windows over our network.

Figure 11 Hearing deficiency detection performance of the proposed TL architecture for two different window lengths.

Figure 11 shows that the proposed TL network with a 2s decision window significantly improves recognition accuracy compared to the 1s decision window analysis. The main goal of this study is to enhance the performance for detecting the hearing condition with a concise decision window, so that we can efficiently use this system in real-life application. For this purpose, first, we analyze the 1s decision window and achieve 91.56% recognition accuracy; still not so high to apply this system in real-life application. Furthermore, to enhance the performance of our proposed diagnosis system, we move on to the 2s decision windows length, and this time we achieve a 5.31% improvement in accuracy compared to the 1s decision window length. In the case of other performance evaluation techniques such as precision, recall, F1 score and Cohen’s kappa, we achieve 5.74%, 3.94%, 5.08%, and 11.02%, improvement, respectively. The improvement indicates the robustness and applicability of our proposed system.

Despite the impressive performance of the proposed system, in some cases, the performance of our network is unsatisfactory. The possible reason for this poorer performance compared to the other successful cases is that in EEG-based BCI application studies, a small SNR and different noise sources are among the greatest challenges. Furthermore, Unwanted signals contained in the main signal can be termed noise, artifact, or interference. Sometimes, the brain may produce some unwanted noise due to the lack of the subject’s proper attention or muscle movement, affecting the detection results. In the experiment, we select concise decision windows (1s and 2s), and working with a short window have many advantages but still very challenging (Geirnaert, Francart & Bertrand, 2020). For these possible reasons, some subjects may provide a lower accuracy compared to the other’s subject (shown in Table 2 and Table 3). Suppose in the 2s decision windows length; if we avoid the two subjects that perform poorer than the other subjects (shown in Table 3), we will achieve 97.62% recognition accuracy. However, the average training and testing accuracy of sixteen subjects with 2s windows length is 100% and 96.87%, respectively, after 100 epochs, whereas the standard deviation is 2.78%.

Furthermore, to study the robustness of the proposed method with a 2s decision window (1s decision windows is not considered in the subsequent analysis), the performance of the proposed model has been compared with other widely used TL architectures. Six popular transfer learning algorithms namely, InceptionResNetV2 (Längkvist, Karlsson & Loutfi, 2014), MobileNet (Pan et al., 2020), ResNet50 (He et al., 2016), VGG16 (Simonyan & Zisserman, 2015), VGG19 (Simonyan & Zisserman, 2015), and Xception (Chollet, 2017) have employed to the time-frequency image of AEP dataset for hearing loss diagnosis. The input size is the same (height- 224* width-224* depth-3) for all the TL architectures. Figure 12 illustrates the performance comparison of six popular TL models with the proposed model. According to Fig. 12, the proposed model achieved higher accuracy compared to the other TL models.

Figure 12 The performance comparison with other pre-trained architectures.

We also reduced the model parameters of VGG16 which help in reducing the model complexity and minimize the computational resources. The total number of all model parameters and performance is represented in Table 4. Table 4 reported that the overall accuracy is less than 61% in all the pre-trained networks, where the models used pre-trained ‘ImageNet’ weights for hearing impairment identification.

Table 4 Performance comparison with six popular TL models.

Pre-network model	Input size	Trainable
parameters	Non-trainable
parameters	Total parameters	Recognition accuracy (%)	
VGG16	224	8,194	134,260,544	134,268,738	57.375	
InceptionResNetV2	224	3,074	54,336,736	54,339,810	54.000	
ResNet50	224	4,098	23,587,712	23,591,810	54.875	
MobileNet	224	2,002	4,253,864	4,255,866	60.250	
Xception	224	4,098	20,861,480	20,865,578	57.625	
VGG19	224	8,194	139,570,240	139,578,434	56.625	
Proposed model	224	103,434,594	9,995,072	113,429,666	96.87 ± 2.78	

In the proposed TL methods (Improved-VGG16), we reduced the total number of parameters of VGG16 (134,268,738 to 113,429,666). Although we reduced the number of parameters, the testing accuracy was still improved to 96.87% from 57.37%. The reason behind the higher accuracy of the proposed model compared to the other TL models is the replacement of some VGG16’s layers with the new layers and fine-tune the higher higher-level parameters, which helps to fit the AEP dataset in the pre-trained network. This replacement consists of adding some dense layers in the fully connected block of VGG16 architecture and adding the dropout layers after every dense layer (shown in Fig. 6). In the fine-tuning block, the time-frequency images are updated with the ‘ImageNet’ weight. This technique helps to fit the dataset in the proposed TL architecture and enhance the overall performance for the hearing loss diagnosis. This experiment is carried out in python, where we used Google colab, Windows 10, Intel(R) Xeon(R) CPU @ 2.30GHz, Tesla K80, and CUDA Version: 10.1.

Discussion

A hearing deficiency detection method based on CWT and improved-VGG16 is proposed in this paper and achieved significantly outperform performance with the shorter decision windows (2s) than the previous state-of-art studies. The proposed improved-VGG16 architecture achieved an average accuracy, precision, recall, f1-score, and Cohen kappa of 96.87%, 96.50%, 97.58%, 97.01%, and 93.74%, respectively.

From Fig. 12, it is clear that our network achieved more than 35% significant improvement compared to the others TL algorithms. In this experiment, we also found a significant effect of the decision window length on the overall performance. We achieved the improvement in the 2s decision window: 5.31% accuracy, 5.74% precision, 3.94% recall, 5.08% in F1 score, and 11.02% Cohen’s kappa than the 1s decision window. The improvement is because the concise decision windows (1s) contain less information and sometimes provide unsatisfactory performance. However, this study aims to build an efficient network that can detect the hearing condition with a concise decision window so that we can able to achieve the decision quickly and can provide more effectiveness in real-life application.

Furthermore, a comparison of the proposed model with existing related studies is represented in Table 5. As seen in Table 5, Hallac et al. (2019) and Dass, Holi & Soundararajan (2016) utilized the convolutional neural network-based classification approach and achieved higher accuracy compared to the other related studies. Hallac et al. (2019) reported that with the raw AEP data and CNN, they achieved 94.1% accuracy. Dass, Holi & Soundararajan (2016) used both the time and frequency domain feature to extract the information from the raw AEP data. They used a feed-forward multilayer network to classify the AEP signal and achieved 90.74% testing accuracy. Both studies achieved a very encouraging performance but need more testing observations to validate the model’s robustness.

Table 5 Performance comparison of related AEP studies.

Reference	Year	Data	Feature extraction	Classification
method	Classification accuracy (%)	
Subject	Class	
(Tang & Lee, 2019)	2019	180	2	WE	TS-PSO	86.17	
(Mahmud et al., 2019)	2019	32	2	Global and
nodal graph	SVM	85.71	
(Dietl & Weiss, 2004)	2004	200	3	WPT	SVM	74.7	
(Zhang et al., 2006)	2006	8	2	DWT	Bayesian
network
classification	78.80	
(Tan et al., 2013)	2013	39	2	SIFT	SVM	87	
(Li et al., 2019)	2019	Observation: 671	2	FFT	SVM	78.7	
(Hallac et al., 2019)	2019	Observation: 671	2	Raw AEP	CNN	94.1	
(Dass, Holi & Soundararajan, 2016)	2016	Observation: 280
Subjects: 151	2	latency, FFT and DWT	A feed-forward multilayer perceptron	90.74	
Proposed work	–	Observations: 3,200	2	CWT	Improved-VGG16	96.87	

In Dietl & Weiss (2004), Mahmud et al. (2019), Tan et al. (2013) and Li et al. (2019), the SVM classifier was used to classify the AEP dataset. Their approach achieved 78.80%, 85.71, 87%, and 78.7% accuracy, respectively. The obtained overall performance is not enough to apply the models in real-life application. Tang & Lee (2019) proposed a TS-PSO hybrid model to classify the two-class AEP dataset. They used Wavelet entropy as a feature extraction method and achieved 86.17% testing accuracy. Zhang et al. (2006) proposed a combination of wavelet analysis and Bayesian networks to classify auditory brainstem response (ABR) signals. For the wavelet analysis, they used the DWT method. Although they conducted an excellent analysis, the overall accuracy is reported 78.80%, which needs improvement.

The experimental outcomes demonstrated that the proposed architecture gain an impressive performance than the other related study for hearing deficiency diagnosis reported in the literature. Although the proposed approach outperforms state-of-art hearing deficiency detection methods, some difficulties are also faced during the experimental analysis. For example, to check the cross-validation and prove the feasibility of our proposed network, a wide range of similar datasets is needed. However, we did not find such dataset for further validation of the proposed method. Another issue is the absence of clear speech envelopes in the dataset. In the previous research, several types of EEG headsets were used to detect the hearing conditions, and these contain a different number of electrodes (1–256). So, the number of electrodes and which electrodes are required to achieve acceptable performance should be determined (Mirkovic et al., 2015; Montoya-Martínez, Bertrand & Francart, 2019; Narayanan & Bertrand, 2018). In most of the studies, the analysis is carried out with ordinary machine learning algorithms, and a few studies are investigated with the deep learning approaches (Krizhevsky, Sutskever & Hinton, 2017; Nossier et al., 2019; Shao et al., 2019). However, most of the studies' testing accuracy is not enough to use the model in real-time as well as real-life applications. A fast and more accurate approach can be an efficient tool for future hearing devices and provide a great application in real-life uses. Our study proposed the time-frequency distribution with a deep learning method and achieved superior performance to other related approaches for hearing loss diagnosis reported in the literature. The key advantages of our proposed method compared to previous studies are written below: Instead of training the AEP dataset with the deep learning architecture from scratch, the proposed study is conducted with a transfer learning strategy, which helps in faster training and better accuracy.

To fit our time-frequency AEP dataset with the pre-trained model weight, we fine-tuned some higher-level parameters where the pre-trained weights are updated with the provided dataset. This strategy helps in enhancing the overall performance for detecting hearing deficiency.

We compare the model’s performance with the six popular TL methods, including VGG16 (Simonyan & Zisserman, 2015), VGG19 (Simonyan & Zisserman, 2015), MobileNet (Pan et al., 2020), ResNet50 (He et al., 2016), InceptionResNetV2 (Längkvist, Karlsson & Loutfi, 2014), and Xception (Chollet, 2017) algorithms where the proposed architecture is superior for hearing deficiency diagnosis.

We also changed some higher-level parameters (after the first layer of the convolutional block five, we remove all the layers and add the new fully connected layer shown in Fig. 5). This approach also helps in reducing the VGG16 parameters and increasing the performance of the proposed improved-VGG16 model.

The proposed approach achieved the height classification accuracy of 96.87%, compared to the previous studies (Ciccarelli et al., 2019; McKearney & MacKinnon, 2019; Ibrahim, Ting & Moghavvemi, 2019; Dietl & Weiss, 2004; Tang & Lee, 2019; Sanjay et al., 2020; Xue et al., 2018; Zhang et al., 2006; Tang & Lee, 2019; Mahmud et al., 2019; Dietl & Weiss, 2004; Zhang et al., 2006; Tan et al., 2013; Li et al., 2019; Hallac et al., 2019; Dass, Holi & Soundararajan, 2016).

The impact of different decision windows is also exhibited in the proposed study, whereas our network provides a significant outcome with a concise decision window.

Conclusions

The proposed hearing loss diagnosis framework consists of two major steps: signal to image transformation and building a hearing deficiency diagnosis system using deep CNN. In the proposed study, the CWT is used to convert the raw signals to time-frequency images. Then, CNN-based improved-VGG16 is used to classify the time-frequency images. This approach achieved better outcomes with fewer trainable parameters, which help to reduce the training time of the model. The applicability and effectiveness of the proposed method are verified by the publicly available AEP dataset, and it achieved 96.87% testing accuracy with a concise decision window. Moreover, this study will help to identify early hearing disorders efficiently. Because of the unstable and subject-specific characteristics of the AEP signal, identification of the AEP signal is challenging. Thus, to enhance the detection system’s accuracy, other AEPs features need to be investigated, and the use of more data variance and conditions can also be improved the outcome.

Supplemental Information

Supplemental Information 1 Code with instruction for the proposed study.

Click here for additional data file.

Additional Information and Declarations

Competing Interests

Author Contributions

Data Availability

The authors declare that they have no competing interests.

Md Nahidul Islam conceived and designed the experiments, performed the experiments, analyzed the data, performed the computation work, prepared figures and/or tables, and approved the final draft.

Norizam Sulaiman conceived and designed the experiments, prepared figures and/or tables, and approved the final draft.

Fahmid Al Farid performed the experiments, authored or reviewed drafts of the paper, and approved the final draft.

Jia Uddin performed the experiments, authored or reviewed drafts of the paper, and approved the final draft.

Salem A. Alyami performed the experiments, authored or reviewed drafts of the paper, and approved the final draft.

Mamunur Rashid conceived and designed the experiments, analyzed the data, performed the computation work, prepared figures and/or tables, and approved the final draft.

Anwar P.P. Abdul Majeed conceived and designed the experiments, prepared figures and/or tables, and approved the final draft.

Mohammad Ali Moni performed the experiments, authored or reviewed drafts of the paper, and approved the final draft.

The following information was supplied regarding data availability:

The code is available in the Supplemental Files.

The data is available at Zenodo: Das, Neetha, Francart, Tom, & Bertrand, Alexander. (2020). Auditory Attention Detection Dataset KULeuven (Version 1.1.0) [Data set]. Zenodo. DOI 10.5281/zenodo.3997352.

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
