# Peer review of "Diagnosis of hearing deficiency using EEG based AEP signals: CWT and improved-VGG16 pipeline"

_PeerJ Computer Science, doi:10.7717/peerj-cs.638_

## Round 0.1 · original submission · Major Revisions

The paper was evaluated by two reviewers who provided thorough comments and suggestions. Both of them find that the paper contains effort to conduct the proposed tasks. The reviewers have given some additional comments and suggestions on how the paper could be improved. Please see their full comments for that. I suggest the authors highlight all modifications in the revised version and answer point-by-point the reviewer's comments. It is also expected that the authors can clearly highlight the feasibility of the experiments.

Reviewer 1 ·

Basic reporting

In this study authors aimed to develop a robust intelligent auditory sensation system utilizing a pre-train deep learning framework by analyzing and evaluating the functional reliability of the hearing based on the AEP response. The proposed method outperformed the state-of-art studies by improving the classification accuracy from 57.375% to 96.87%. The subject of the paper is interesting and hopefully it will be eventually useful.

Experimental design

No comment.

Validity of the findings

No comment.

Additional comments

Some revisions are as follows:
(1) In line 179, please add the units of “224*224”.
(2) In line 211, please explain why the raw signal was converted into a time-frequency image using a “2-second” time window.
(3) In equation 3, please the explain the “b”, “c”, “t” and “r”.
(4) In line 334-340, please explain why the new dense layers and dropout layer were added in VGG16, and how to select the parameters like dropout value, batch size and learning rate. Or authors should compare different architectures and parameters to test model performance.
(5) In Discussion, the content of algorithms comparisons should be moved in “Result of the Experiment and Analysis”.
(6) There are some typos like “This operation” in line 239.

Reviewer 2 ·

Basic reporting

a. the article must be clearly written and use unambiguous technically correct text. Several of them have been awarded in the attachment. Please modify other paragraphs that are easy to be ambiguous.
b. the article should include sufficient introduction and background to demonstrate how the work fits into the broader field of knowledge. Relevant prior literature should be appropriately referenced. There are already many commercial products, and their problems have not appeared in this article.

Experimental design

No.

Validity of the findings

The experimental results obtained by the author are higher than those of previous studies.
However, the author's experimental cases are too few to prove the feasibility scientifically, which needs to be highlighted in the Discussion part.

Additional comments

Scientific research on hearing impairment is of great practical significance. Early hearing screening is helpful to improve the problem. The study of this paper aims to propose a robust intelligent auditory sensation system utilizing a pre-train deep learning framework by analyzing and evaluating the functional reliability of the hearing based on the AEP response. The experimental results obtained by the author are higher than those of previous studies.However, the author's experimental cases are too few to prove the feasibility scientifically, which needs to be highlighted in the Discussion part.

Annotated reviews are not available for download in order to protect the identity of reviewers who chose to remain anonymous.

---

## Round 0.2 · accepted · Accept

The manuscript has been re-evaluated by two reviewers. Both reviewers have suggested that it has been well-revised and all the comments have been addressed accordingly.

Reviewer 1 ·

Basic reporting

No comment.

Experimental design

No comment.

Validity of the findings

No comment.

Additional comments

The authors have modified their work and satisfied my previous comments. Therefore, from my point of view, this paper should be accepted to be published.

Reviewer 2 ·

Basic reporting

no comment

Experimental design

no comment

Validity of the findings

no comment

Additional comments

No further comments.